# Prostate Cancer Diagnostic Algorithm as a “Road Map” from the First Stratification of the Patient to the Final Treatment Decision

**DOI:** 10.3390/life11040324

**Published:** 2021-04-07

**Authors:** Hana Sedláčková, Olga Dolejšová, Milan Hora, Jiří Ferda, Ondřej Hes, Ondřej Topolčan, Radka Fuchsová, Radek Kučera

**Affiliations:** 1Department of Urology, Faculty of Medicine in Pilsen, University Hospital, 305 99 Pilsen, Czech Republic; sedlackovah@fnplzen.cz (H.S.); dolejsovao@fnplzen.cz (O.D.); horam@fnplzen.cz (M.H.); 2Department of Medical Imaging, Faculty of Medicine in Pilsen, University Hospital, 304 60 Pilsen, Czech Republic; ferda@fnplzen.cz; 3Department of Pathology, Faculty of Medicine in Pilsen, University Hospital, 305 99 Pilsen, Czech Republic; HES@fnplzen.cz; 4Department of Immunochemistry Diagnostics, Faculty of Medicine in Pilsen, University Hospital, 305 99 Pilsen, Czech Republic; topolcan@fnplzen.cz (O.T.); fuchsovar@fnplzen.cz (R.F.); 5Institute of Pharmacology and Toxicology, Medical Faculty in Pilsen, Charles University, 304 60 Pilsen, Czech Republic

**Keywords:** prostate cancer, diagnostic algorithm, prostate health index, biopsy, Gleason score, magnetic resonance, positron emission tomography, diagnosis, imaging, prostate-specific membrane antigen

## Abstract

The diagnostics of prostate cancer are currently based on three pillars: prostate biomarker panel, imaging techniques, and histological verification. This paper presents a diagnostic algorithm that can serve as a “road map”: from initial patient stratification to the final decision regarding treatment. The algorithm is based on a review of the current literature combined with our own experience. Diagnostic algorithms are a feature of an advanced healthcare system in which all steps are consciously coordinated and optimized to ensure the proper individualization of the treatment process. The prostate cancer diagnostic algorithm was created using the prostate specific antigen and in particular the Prostate Health Index in the first line of patient stratification. It then continued on the diagnostic pathway via imaging techniques, biopsy, or active surveillance, and then on to the treatment decision itself. In conclusion, the prostate cancer diagnostic algorithm presented here is a functional tool for initial patient stratification, comprehensive staging, and aggressiveness assessment. Above all, emphasis is placed on the use of the Prostate Health Index (PHI) in the first stratification of the patients as a predictor of aggressiveness and clinical stage of prostrate cancer (PCa). The inclusion of PHI in the algorithm significantly increases the accuracy and speed of the diagnostic procedure and allows to choose the optimal pathway just from the beginning. The use of advanced diagnostic techniques allows us to move towards to a more advanced level of cancer care. This diagnostics algorithm has become a standard of care in our hospital. The algorithm is continuously validated and modified based on our results.

## 1. Introduction

Prostate cancer (PCa) is the most frequent malignant disease to occur in men. According to The International Agency for Research on Cancer (IARC), 1,414,259 new cases of PCa were reported and 375,304 men died of PCa worldwide in 2020 [1]. PCa’s incidence and mortality are connected to the human development index (HDI). The disease is most prevalent in developed countries, while its mortality rate is highest in low-HDI countries [2]. The risk of PCa increases with age. The majority of PCa cases are diagnosed in men older than 65 [3]. PCa is a highly heterogeneous disease, ranging from a clinically insignificant manifestation that requires only active surveillance, to a highly aggressive castration-resistant type of tumor that requires a quick and radical course of action. Differences in the incidence rate of PCa worldwide primarily reflect differences in the use of diagnostic testing. Accurate diagnostics and PCa staging are imperative for the selection of the most appropriate therapeutic strategy [4].

The diagnostics of PCa are currently based on three pillars: prostate biomarker panel, imaging techniques, and histological verification. This paper presents a diagnostic algorithm that can serve as a “road map” delineating the course of treatment: from initial patient stratification to the final decision regarding treatment. The algorithm is based on a review of the current literature combined with our own experience.

## 2. Diagnostic Algorithm

### 2.1. PCa Diagnostic Algorithm–A Tool for Patient Stratification, Staging and Aggressiveness Assessment

The first algorithm was created a few years ago. Since then, the algorithm has been supplemented every year with new knowledge and new diagnostic procedures introduced into clinical practice. This was done to ensure that the algorithm continues to reflect the most current procedures that are applied in our university hospital.

Diagnostic algorithms are a feature of an advanced healthcare system in which all steps are consciously coordinated and optimized to ensure the proper individualization of the treatment process. The PCa diagnostic algorithm was created using the prostate specific antigen (PSA) and in particular the Prostate Health Index (PHI) in the first line of patient stratification. It then continued on the diagnostic pathway via imaging techniques, biopsy, or active surveillance, and then on to the treatment decision itself (Figure 1).

The first step is to have patients stratified into three groups according to PSA and PHI levels. If the PSA and PHI levels are low, patients are rated as benign: they will still be monitored and will be tested again, usually after six months.

If the level of PSA is above the reference ranges for the patient’s age group, and/or the PHI level is over 40, the second step is to perform imaging techniques. We use multi-parametric magnetic resonance imaging (mpMRI) to localize the lesion, but also to evaluate a more detailed anatomy before surgery. We performed ^68^Ga-PSMA-11 PET/MRI as part of the comprehensive staging in a selected group of patients before radical prostatectomy, as well as in primary diagnostics before histological verification. This is done in cases where there is a strong suspicion that the patient has a high-risk of developing PCa, or has locally advanced PCa and the extensive staging leads to a change in treatment management. In the middle of the algorithm, the current status of the PCa is proved using biopsy, in order to achieve that each patient with suspected PCa undergoes an mpMRI.

The biopsy holds a key position located in the middle of the algorithm. In order to achieve the best results in histological verification of significant PCa, we perform an MRI/transrectal ultrasound-guided (TRUS) fusion software-based targeted biopsy of the prostate. If the man is biopsy naïve, an additional systematic biopsy will be performed to determine the extent of the tumor and to help in planning the surgery–nerve sparing or not.

Finally, at the end of the algorithm, a treatment decision is made. Based on the results of histology, evaluated using the Gleason score and the imaging examinations, the appropriate method of treatment is selected according to the stage of the disease. As a relatively new approach, active surveillance is also incorporated into the algorithm.

### 2.2. The Algorithm Is Based on Our Experience, Results and Knowledge

It has been our experience that the value of PHI level can be used for validation in patients after radical prostatectomy. A total of 787 patients were examined and subsequently operated from 1/2013 to 12/2019. A definitive Gleason score was determined. PHI values were compared with definitive staging and grading. The study confirmed a very good ability of PHI to distinguish GS < 7 (low aggressiveness) and GS ≥ 7 (higher aggressiveness) prostate tumors and thus, PHI was added to the first line of biochemical assessment of the tumor aggressiveness [5].

We have performed 3T mpMRI to detect PCa lesions as a standard method from 2012. This step also decreases the over diagnosis of PCa. From 1/2018 to 2/2020, 150 patients underwent ^68^Ga-PSMA-11 PET/MRI as part of comprehensive staging; this examination is the only one under the clinical trial in our country.

Magnetic resonance imaging and targeted biopsy (MRI/TBx) were performed from 1/2017 to 12/2019 in the examination of 450 patients.

### 2.3. PHI as a Tool for the First (Initial) Stratification of the Patient

PCa diagnostics using biomarkers started in the 1980s with the total PSA (tPSA) measurement. Total PSA has a limited sensitivity and specificity for PCa detection [6]. Seeking better sensitivity and specificity, free PSA (fPSA) was introduced and then (2])proPSA. These developments enabled physicians not only to start using biomarkers, but also to calculate parameters; namely, the percentage of fPSA (%freePSA = (fPSA/tPSA) * 100) and Prostate Health Index, PHI (PHI = (([-2])proPSA/fPSA) × √tPSA). These parameters, PHI especially, contributed to PCa aggressiveness assessment using biochemical methods [7]. One of the largest recent studies carried out by PROMETHEUS, a Multicentric European Study, confirmed PHI as one of the strongest predictors of PCa, correlating with the Gleason Score (GS). In our own studies, firstly monocentric [5,8] and later on multicentric (with our partners) [9,10], we proved the PHI’s ability to distinguish between PCa GS < 7 (low aggressiveness) and GS ≥ 7 (higher aggressiveness).

### 2.4. The Key Role of Imaging Techniques in Staging and Surgical Navigation

A pathway with mpMRI combining T2-weighted, dynamic contrast-enhanced (DCE) and diffusion weighted imaging (DWI) has been shown to be accurate in significant PCa. Prostate anatomy is best assessed by T1 and T2 weighted images, with the DCE and DWI contributing functional information. There is also evidence that mpMRI tends to detect higher risk disease, which makes it attractive as a potential triage test [11]. mpMRI in the diagnostics of PCa is very often used for its high sensitivity and specificity. Sensitivity increases especially with tumor size and aggressiveness. The results are excellent, especially for significant tumors: tumor volume ≥ 0.5 mL or GS ≥ 7 [12]. Imaging with mpMRI plays two roles in PCa diagnostics. Firstly, it functions as a secondary screening test, exempting men with nonsuspicious tests from biopsy. MRI reduced the need for biopsy by 68% in men with PSA 3.0 µg/L. The second function of MRI is to provide an image of the lesion(s), so that sampling can be more precise [11].

As part of the unification of the MRI description, the European Society for Uroradiology (ESUR) introduced the PI-RADS (Prostate imagining reporting and data system) classification system. This prostate sector diagram employs forty-one sectors/regions: thirty-eight for the prostate, two for the seminal vesicles and one for the external urethral sphincter [13,14].

PET/CT with radiolabeled choline analogs is widely used in clinical practice for prostate cancer staging. ^18^F-fluoroethylcholine PET demonstrated higher accuracy than MRI for the detection of primary prostate cancer; specificity was however limited by choline uptake in benign lesions [15]. Since 2012, [^18^F]- and [^68^Ga]-labeled inhibitors of prostate-specific membrane antigen (PSMA) entered early clinical development for PET imaging of PCa and showed immediate promise for sensitive and specific identification of local and distant sites of disease [16,17]. Results from [^68^Ga]-PSMA-11 PET/MRI and PET/CT in Figure 2, Figure 3 and Figure 4. To summarize, according to the available systematic reviews and clinical trials, the sensitivity and specificity in primary staging of PCa using PSMA ligands is usually above 40% and over 85%, respectively. The impact on therapy planning was also investigated, performing PET/CT or, less frequently, PET/MRI using PSMA ligands, the therapeutic procedure changes in approximately 21% of patients in the primary staging [18,19].

### 2.5. The Basic Role of the Biopsy in Tumor Aggressiveness Assessment

A necessary condition for the initiation of PCa therapy is PCa histological verification using biopsy. The transrectal ultrasound (TRUS) navigated biopsy is used as a basic procedure [20]. The second option is the cognitive biopsy in which the result of the imaging technique, most often mpMRI, is known. Currently, the preferred procedure is the fusion biopsy, where images from mpMRI and TRUS are merged by software [20,21]. MRI information can be used to guide prostate biopsy cores, especially MRI/TRUS fusion software-based targeted biopsy of the prostate (MRI-TBx) to suspicious areas in the prostate. MRI-TBx has a higher detection rate for significant PCa and a lower detection rate for insignificant PCa compared with T-Bx [22]. Nonetheless, some lesions might also be missed on MRI-guided biopsies and these are the patients who pose a diagnostic challenge. With the introduction of ^68^Ga-PSMA, ligands which exhibit almost exclusive expression in the prostate and increased expression in PCa are more often detected [23]. PSMA-PET/MRI in combination with a newly developed fusion biopsy system-PET/TRUS and PET/MRI/TRUS fusion-proved to be a valuable tool for the detection of PCa in patients following a prior negative prostate biopsy and is therefore attracting increasing attention [24].

Based on biopsies, the Gleason score (GS) has been used since the 1960s as the main grading system for PCa cell assessment. The GS ranges from 1 to 10 and was considered a main factor when a treatment plan was determined. This was the case until 2016 when the International Society of Urological Pathology (ISUP) revised the PCa grading system and a new scale, the 5 ISUP Grades, was established [25].

### 2.6. Active Surveillance–A Suitable Procedure for Tumors with Low Aggressiveness

With more and more advanced diagnostic methods and increasingly accurate assessment of tumor aggressiveness, new approaches such as active surveillance can be applied instead of urgent surgery [26]. Active surveillance is an excellent example of how the medical paradigm has slowly changed during recent years. As aggressiveness is the main predictive factor for subsequent treatment management in the case of PCa, the main current task is to make precise and timely aggressiveness assessments [27]. Considering the side effects of radical prostatectomy (incontinence or sexual dysfunction), which is indicated in the case of highly aggressive PCa (GS ≥ 7), active surveillance seems to be the suitable option for PCa with low aggressiveness (GS < 7).

## 3. Discussion

The above-described PCa diagnostic algorithm has a few limitations. One major difficulty may arise when using highly specific imaging methods with the latest radiotracers; these, however, are not widely available. Nonetheless, the algorithm was designed precisely with the aim of incorporating these state-of-the-art diagnostic methods and implementing them in clinical practice in order to achieve a clear indication.

Hybrid imaging using PET and MRI has been intentionally incorporated into the algorithm. MRI is perfect for the imaging of both the prostate, especially for targeted prostate biopsy, and for the detection of lymph node metastases. It has an irreplaceable role in preoperative lymph node staging. Having used the [^68^Ga]-PSMA-11 as the latest radiotracer with very promising results, we believe that thus performed staging is highly specialized and yields the best results. [^68^Ga]-labelled PSMA ligand could be superior to choline tracers in its ability to obtain high contrast. PSMA tracer can detect lesions characteristic of PCa with improved contrast when compared to the standard [^18^F]-fluoromethylcholine, especially at low PSA levels. A significant advantage of [^68^Ga]-PSMA-11 is that lesions characteristic of lymph node metastases are frequently presented in very high contrast when compared to choline. The superior contrast in [^68^Ga]-PSMA-11 has also been demonstrated in most skeletal metastases [28].

Due to the low availability of PET/MRI scanners, PET/CT can be used instead. When this is the case, however, we lose the possibility of using images for targeted prostate biopsy and we are forced to perform further examinations in the form of at least mpMRI of the prostate, which delays further treatment decisions and initiations. Furthermore, the hybrid imaging method PET/MRI has the advantage of a reduced radiation dose compared to PET/CT.

Even though the PSMA-PET scan has shown considerable early promise, its availability is limited and incurs considerable cost. Furthermore, since prostate cancer patients commonly undergo mpMRI of the prostate, there is the possibility of a one-stop staging modality in the form of a whole-body MRI (wb-MRI). According to EAU guidelines, wb-MRI is more sensitive than conventional imaging methods and is more sensitive than choline PET/CT in its detection of bone metastases. Nevertheless, choline PET/CT had the highest specificity for diagnostic evaluation [29]. Wb-MRI with DWI is an effective method for overall staging in PCa, as it can detect metastases in normal-sized lymph nodes and early intramedullary bone metastases before the appearance of cortical destruction or reactive processes [30]. The LOCATE trial designed to compare the detection of prostate cancer using conventional imaging methods with wb-MRI will certainly yield promising results [31]. Due to the fact that a whole-body MRI is not a standard imaging method in our hospital, this alternative is not applicable in our case. Currently, mpMRI is the standard method for prostate imaging and it plays an important role in the detection, targeted biopsy, local staging, and risk classification of prostate cancers. Many studies have compared the bi-parametric MRI imaging protocol consisting of T2-weighted imaging and DWI with a standard multi-parametric imaging protocol for the detection of PCa. There is no significant difference regarding the detection of PCa [32,33,34].

Our hospital is committed to performing mpMRI. This is based on knowledge gained from experience: DCE can in some cases help detect prostate cancer in both PZ and TZ. It is sometimes referred to as a “backup” sequence, especially if DWI/ADC is degraded by artifacts [35]. In PI-RADS version 2.1, DCE is used to differentiate scores of 3 and 4 in the peripheral zone. If we have a DWI score of 3 and early saturation is present, the finding is upgraded to a score of 4, which may help achieve a more accurate aggressiveness classification and individualized treatment of prostate cancer [13].

## 4. Conclusions

The PCa diagnostic algorithm presented here is a functional tool for initial patient stratification, comprehensive staging and aggressiveness assessment.

The use of advanced diagnostic techniques allows us to move towards to a more advanced level of cancer care that is more beneficial for patients. This diagnostics algorithm has become a standard of care in our hospital. The algorithm is continuously validated and modified based on our results.

## Figures and Tables

**Figure 1 life-11-00324-f001:**
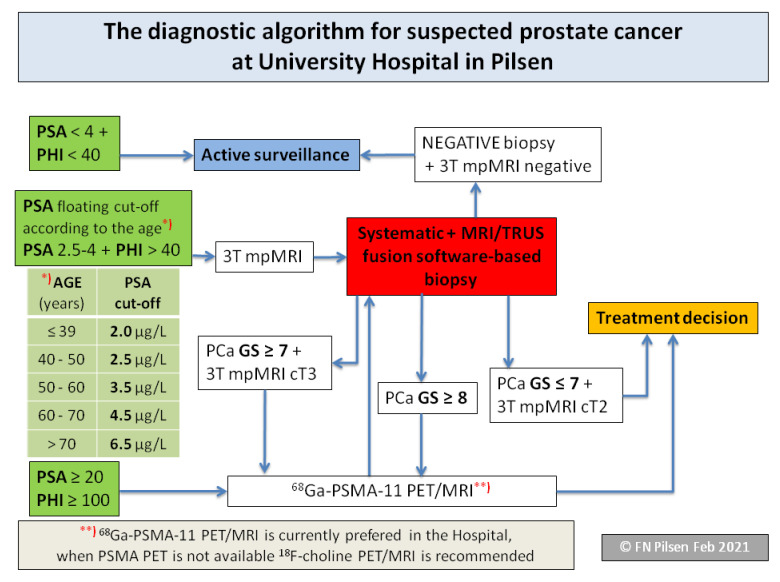
Prostate cancer diagnostic algorithm.

**Figure 2 life-11-00324-f002:**
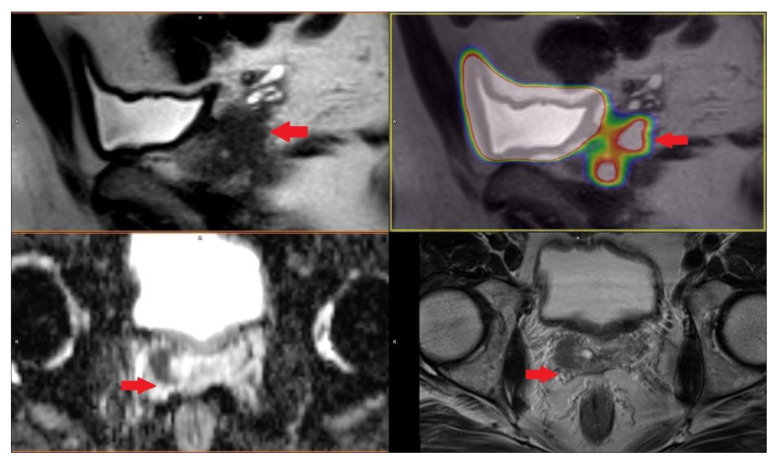
Patient with prostate cancer GS 7 (4 + 3) in right lobe with right seminal vesicle invasion cT3b iPSA 4.17 PHI 64.93 in [^68^Ga]-PSMA-11 PET/MRI.

**Figure 3 life-11-00324-f003:**
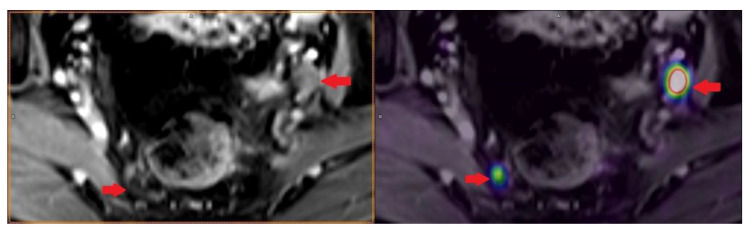
Patient with lymph nodes metastases in prostate cancer GS 9 (4 + 5) iPSA 22.85 PHI 130.63 in [^68^Ga]-PSMA-11 PET/MRI.

**Figure 4 life-11-00324-f004:**
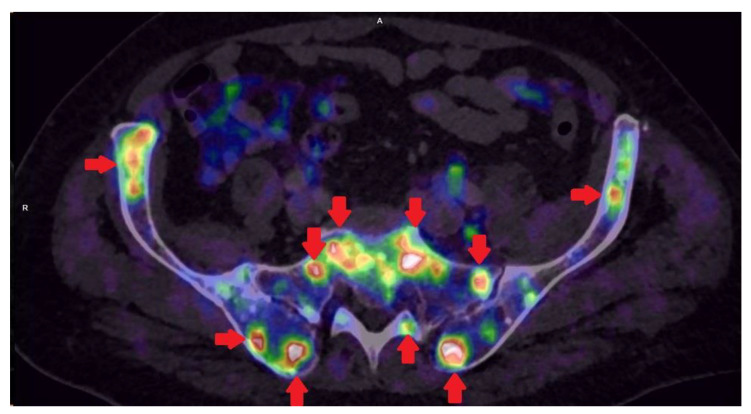
Patient with multiple bones metastases in prostate cancer GS 8 (4 + 4) iPSA 45.33 PHI 176.8 in [^68^Ga]-PSMA-11 PET/CT.

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
