# Peer review of "Prostate Cancer Diagnostic Algorithm as a “Road Map” from the First Stratification of the Patient to the Final Treatment Decision"

_life, 2021, doi:10.3390/life11040324_

Round 1

Reviewer 1 Report

The present paper is overall well written and interesting. I only have some minor comments:

1) 68Ga-PSMA-11 PET/MRI can be considered the imaging gold standard for advanced prostate cancer. However, both the tracer and hybrid PET/MRI scanners are not widely available. Do the Authors believe that the algorithm would still work well with 18F-fluoroethylcholine PET/CT? What about whole body MRI? The latter should at least be mentioned in the paper I believe.

2) What about abbreviated MRI protocols? mpMRI is amazing, but bpMRI is good too and could reduce costs with similar diagnostic accuracy (10.2214/AJR.20.23219)

3) Page 3 line 124: MR spectroscopy has been out of the game for long now and does not deserve to be mentioned.

4) A second figure showing some nice PET/MRI images with pathological confirmation would be a nice addition. 

Author Response

Dear reviewer, thank you for your interesting questions. Based on the comments, we decided to add a Discussion section. In it we explained our opinions and answered the reviewers’ questions in detail.
Please see the attachment.

Reviewer 2 Report

The autors present a diagnostic algorithm that basically respresents common staging as advised by international guidelines, except for the inclusion of PHI, which is an individual option.

Author Response

Yes, based on the international guidelines the algorithm was created. However, included into this algorithm our long term experience with PHI. We believe that it could be that sharing our experiences can be beneficial for readers.

Reviewer 3 Report

The article "Prostate Cancer Diagnostic Algorithm as a “Road Map” from the First Stratification of the Patient to the Final Treatment Decision" authored by Sedláčková et al is a good and comprehensive study.

Author Response

Dear reviewer, thank you for your comment.